# Assessing the Genotoxicity of Cellulose Nanomaterials in a Co-Culture of Human Lung Epithelial Cells and Monocyte-Derived Macrophages

**DOI:** 10.3390/bioengineering10080986

**Published:** 2023-08-21

**Authors:** Célia Ventura, Fátima Pinto, Ana Filipa Lourenço, Jorge F. S. Pedrosa, Susete N. Fernandes, Rafaela R. da Rosa, Maria Helena Godinho, Paulo J. T. Ferreira, Henriqueta Louro, Maria João Silva

**Affiliations:** 1Department of Human Genetics, Instituto Nacional de Saúde Doutor Ricardo Jorge, Av Padre Cruz, 1649-016 Lisbon, Portugal; celia.ventura@insa.min-saude.pt (C.V.); fatimapinto31081@gmail.com (F.P.); henriqueta.louro@insa.min-saude.pt (H.L.); 2ToxOmics—Centre for Toxicogenomics and Human Health, NOVA Medical School, NOVA University Lisbon, 1169-056 Lisbon, Portugal; 3RAIZ—Forest and Paper Research Institute, 3800-783 Aveiro, Portugal; ana.filipa@thenavigatorcompany.com; 4University of Coimbra, CIEPQPF, Department of Chemical Engineering, Pólo II, R. Sílvio Lima, 3030-790 Coimbra, Portugal; jorge_fsp@live.com.pt (J.F.S.P.); paulo@eq.uc.pt (P.J.T.F.); 5i3N/CENIMAT, Department of Materials Science, NOVA School of Science and Technology, NOVA University Lisbon, Campus de Caparica, 2829-516 Lisbon, Portugal; sm.fernandes@fct.unl.pt (S.N.F.); rr.rosa@fct.unl.pt (R.R.d.R.); mhg@fct.unl.pt (M.H.G.)

**Keywords:** nanofibrillated cellulose, nanocrystalline cellulose, safety assessment, biocompatibility, respiratory effects

## Abstract

Cellulose micro/nanomaterials (CMNMs) are innovative materials with a wide spectrum of industrial and biomedical applications. Although cellulose has been recognized as a safe material, the unique properties of its nanosized forms have raised concerns about their safety for human health. Genotoxicity is an endpoint that must be assessed to ensure that no carcinogenic risks are associated with exposure to nanomaterials. In this study, we evaluated the genotoxicity of two types of cellulose micro/nanofibrils (CMF and CNF) and one sample of cellulose nanocrystals (CNC), obtained from industrial bleached *Eucalyptus globulus* kraft pulp. For that, we exposed co-cultures of human alveolar epithelial A549 cells and THP-1 monocyte-derived macrophages to a concentration range of each CMNM and used the micronucleus (MN) and comet assays. Our results showed that only the lowest concentrations of the CMF sample were able to induce DNA strand breaks (FPG-comet assay). However, none of the three CMNMs produced significant chromosomal alterations (MN assay). These findings, together with results from previous in vitro studies using monocultures of A549 cells, indicate that the tested CNF and CNC are not genotoxic under the conditions tested, while the CMF display a low genotoxic potential.

## 1. Introduction

Nanocelluloses or cellulose micro/nanomaterials (CMNMs) are remarkable eco-friendly cellulose-derived materials displaying at least one dimension below 100 nanometers [1,2]. While keeping the cellulose properties, CMNMs benefit from the advantageous features of nanomaterials, thus opening opportunities for innovation in the fields of materials science and industry [3]. Nanocelluloses can be categorized, according to size and preparation method, into three main categories: cellulose micro- or nanofibrils (CMF or CNF, also called micro/nanofibrillated cellulose), cellulose nanocrystals (CNC, also named nanocrystalline cellulose) and bacterial nanocellulose (BNC, also named microbial cellulose or biocellulose) [4,5]. Despite the composition similarity, the diverse CMNMs may differ in one or several physicochemical properties, such as morphology, size or crystallinity, depending on the sources from which the cellulose is obtained and the extraction methods applied [2,6,7]. CMF and CNF have been produced from several sources, including wood, cotton and other non-wood plants, grasses and even pulps from agricultural wastes [8,9]. To achieve cellulose defibrillation, cellulose-enriched materials are subjected to mechanical processes by passing through homogenizers, micro-fluidizers or grinders [10,11]. Those processes are usually facilitated by a chemical pre-treatment of the cellulose materials through cationization, carboxymethylation or 2,2,6,6-Tetramethylpiperidine-1-oxyl radical (TEMPO)-mediated oxidation [12]. A more recent approach that facilitates cellulose fibrillation is an enzymatic treatment [10] that has received increased attention due to its reduced energy consumption, making it more environmentally friendly [1,13]. Due to their high versatility, CMNMs are promising materials for innovative medical applications, including tissue engineering, biomedical implants, drug delivery, wound healing or antimicrobial treatment strategies [14,15]. Other potential applications of these nanomaterials have comprised areas as diverse as food technology, paintings, cosmetics, automotive materials, packaging and gas barrier films [16].

Cellulose has been generally considered as a safe and biocompatible material because of its natural origin. In addition, due to its indigestibility by the human gastrointestinal system, it has even been used in food packaging or as a food additive (as thickener or stabilizer) [17,18]. However, at the nanoscale, the high aspect ratio of CMF/CNF and their biodurability in the human lungs [19] resemble characteristics of hazardous nanofibers (e.g., carbon nanotubes, CNT). These properties raised concerns about their potential to cause adverse effects, particularly in the respiratory system of occupationally exposed individuals. Also worth considering was the previous recognition that properties inherent to nanosized materials, such as their morphology, degree of crystallinity, surface chemistry (zeta potential), colloidal stability and aggregation properties, among others, may convert non-toxic raw or micromaterials into toxic nanomaterials [20]. Thus, it is crucial to assess the potential hazard of newly produced CMNMs, particularly their genotoxicity, an adverse effect in close relationship with carcinogenicity [17,21]. Recently, the toxicological properties of several CMNMs have been examined in vitro and in vivo [22,23] but no consensus about their safety has been reached yet. Some studies showed a low internalization ability in cells, and no strong evidence of significant cytotoxicity or genotoxicity [24]. However, some contradictory results have also been reported, pointing to in vivo [25,26] and in vitro [21,27,28] genotoxicity of some CNF, and inflammatory effects of several CNC in macrophages that ranged from mild to severe, according to their functionalization [29,30,31,32].

Animal models have been very useful in the investigation of the health effects of inhaled substances, including nanomaterials, but have failed to mimic some human responses [33,34]. Moreover, the increasing effort that has been made towards the reduction of animal testing in response to ethical concerns has promoted the development of alternative experimental systems to assess the adverse effects of inhaled toxicants [35]. Even though in vitro cell cultures have been envisaged as suitable alternatives to animal testing, there is still a need to implement more complex models, such as co-culture systems, to more closely resemble the lung environment [33]. Co-cultures have proved their usefulness in investigating cell-to-cell interactions, bio-nano interactions, and in mechanistic studies of drugs’ action and targets [36,37,38,39]. In vitro toxicological inhalation studies usually rely on lung-derived cellular models, given that epithelial cells cover the lung surface forming the first layer exposed to inhaled particles [33]. It is also known that the immune response plays a central role in the body protection against external substances. On the other hand, in vivo genotoxicity studies have shown that when nanomaterials are able to induce permanent genetic damage, this effect is frequently associated with a persistent inflammatory reaction that secondarily produces genotoxicity via reactive oxygen species generation [40,41,42]. Thus, the application of co-culture models that include immune cells is believed to provide data that better reflect the effects in the human tissues than standard monoculture models [43]. In this study, a co-culture of human A549 cells and THP-1 monocytes differentiated into macrophages was used, a model that had previously proved its adequacy to study the genotoxicity of multiwalled CNT [39]. A549 cells are human alveolar type II epithelial cells, whereas differentiated THP-1 cells have been considered surrogates of alveolar macrophages, which are involved in the lung immune response.

The aim of the present work was to assess the genotoxic effects of different CMNMs produced from industrially bleached *Eucalyptus globulus* kraft pulp, using a co-culture system of human lung epithelial cells and monocyte-derived macrophages. The combination of the comet and the micronucleus (MN) assays was intended to provide information on several endpoints related to cytotoxicity, genotoxicity and oxidative damage [44]. Indeed, the comet assay detects DNA damage/strand breaks and oxidant damage to DNA (FPG-comet assay) with high sensitivity, whereas the MN assay identifies chromosomal breaks or loss, thus characterizing biological events that are linked to cell transformation and cancer [44,45,46]. Due to the close association between genotoxicity and carcinogenicity, the assessment of the genotoxic effects of these CMNMs is pivotal to identify potential risks from human exposure to these materials before they enter the market.

## 2. Materials and Methods

### 2.1. Synthesis and Characterization of Cellulose Nanofibrils and Nanocrystals

CNF, CMF and CNC were obtained from industrial bleached *Eucalyptus globulus* kraft pulp (BEKP) (80–85 wt% cellulose, 14–19 wt% xylan, 0.3 wt% lignin and 0.4 wt % extractives) and were fully characterized for the fibrillation yield, carboxyl content (C_COOH_), degree of polymerization (DP) and intrinsic viscosity ([η]) as described in Pinto et al., 2022 [47]. Briefly, for the preparation of the fibrillated celluloses, the BEKP was first refined in a PFI beater. The CNF was obtained by subjecting the refined fibers to a TEMPO- mediated oxidation, by addition of 0.016 g of radical TEMPO, 0.1 g of NaBr and 5 mM of NaClO per gram of fibers. The reaction took place for around 2 h. For the CMF preparation, the refined fibers were enzymatically treated by addition of an enzymatic cocktail (10% endocellulase, 10% exocellulase, and 5% hemicellulase) at a dosage of 300 g/ton of fiber. The hydrolysis took place at 50 °C for 2 h. After the chemical and enzymatic treatments, both samples were properly washed with distilled water and subsequently subjected to a mechanical treatment in a high-pressure homogenizer (GEA Niro Soavi, model Panther NS3006 L, GEA Group Aktiengesellschaft, Düsseldorf, Germany) with 2 passages, the first one at 500 bar and the second at 1000 bar. The CNC were produced by chemical hydrolysis with sulfuric acid (62 wt%) at acid solution/fibers ratio of 8:1. The mixture was kept at 55 °C for 75 min. The cellulose nanocrystals were recovered and purified by centrifugation and dialysis against ultrapure water. The morphology, hydrodynamic diameter (z-Average) and surface charge of the three CMNMs samples were also analyzed when dispersed in phosphate buffered saline (PBS) and in complete RPMI 1640 (Thermo Fisher Scientific, Waltham, MA, USA) culture medium [47]. The morphology and estimated diameter were analyzed by Transmission Electron Microscope (TEM) imaging using the negative staining technique [47]. Representative TEM images of the three CMNMs diluted in PBS are shown in Figure 1. The most relevant properties for the possible CNMN genotoxicity are presented in Table 1. It is worth to note that the yield of fibrillation of CMF-ENZ sample, as determined by gravimetric analysis, is apparently quite low (4.9%). Although gravimetric analysis has been considered as a simple and fast method to compare the effect of distinct treatment combinations in the overall level of degradation of the fibers, the measured yield of fibrillation (non-sedimentable material) might be affected by the size of the fibrils, their charge (effects of charge stabilization) and the number of fibrils that attach to the initial fiber fragments’ body. In the specific case of the CMF-ENZ, the low level of carboxylic groups (that can act as stabilizers in the CNF, resulting in higher values of yield of fibrillation) are not enough to keep the longer fibrils in suspension, allowing them to settle. It was not possible to determine the diameter of CNF-TEMPO dispersed in culture medium due to the presence of proteins that masked the nanofibrils.

A stock suspension at 1.5 mg/mL was prepared for the three CMNMs by dispersing the CNF-TEMPO and CMF-ENZ gels and CNC powder in PBS for 30 min, using a magnetic stirrer. The suspensions were diluted in complete cell culture medium to obtain the concentrations tested.

### 2.2. Cell Culture and Exposure to Nanofibers

The human lung carcinoma epithelial (A549) cell line was obtained from the American Type Culture Collection (ATCC, Manassas, VA, USA, CCL-185). A549 cells and the human monocytic leukaemia (THP-1) cell line (ATCC, TIB-202) were both cultured in complete culture medium (CM) consisting of RPMI 1640 medium (Thermo Fisher Scientific, Waltham, MA USA) supplemented with 10% heat-inactivated fetal bovine serum (FBSi) (Thermo Fisher Scientific), 1% penicillin/streptomycin (1.000 U/mL penicillin and 10 mg/mL streptomycin; Thermo Fisher Scientific) and 1% fungizone (0.25 mg/mL; Thermo Fisher Scientific), at 37 °C and 5% CO_2_. THP-1 monocytes were cultured on transwell inserts with a nominal pore size of 0.4 μm (Greiner Bio-One GmbH, Kremsmünster, Austria) at a density of 0.2 × 10^5^ cells/mL, and were differentiated into macrophages by 48 h incubation with 100 ng/mL of 12-O-tetradecanoylphorbol-13-acetate (TPA, Sigma-Aldrich, St. Louis, MO, USA), followed by 48 h incubation in serum-free medium. Log-phase A549 cells were plated on 12-well plates at a density of 0.5 × 10^5^ cells/mL and cultured for 24 h. Then, the inserts with differentiated THP-1 cells were placed directly on the wells with the A549 cells and the resulting co-culture was incubated for further 24 h in complete medium. Semi-confluent cell cultures were exposed to 1.5, 3, 6, 12.5, 25 and 50 μg/cm^2^ of each CMNM and maintained at 37 °C in 5% CO_2_. To ensure that THP-1 and A549 cells were exposed to the same CMNM concentrations, the dispersions were added to the apical and basolateral sides of the insert.

### 2.3. Cytokinesis-Blocked Micronucleus (CBMN) Assay

The CBMN assay was performed according to the OECD 487 guideline [48] and adapted to overcome the interference of NMs [49]. Briefly, A549/THP-1 cells grown in co-culture were exposed to each CMNM for 24 h. After that period, cytochalasin B (6 μg/mL, Sigma-Aldrich) was added to each well and the cell culture was incubated for another 24 h. Negative (non-treated cells) and positive (50 μg/mL mitomycin C, Sigma-Aldrich) controls were included for each experiment. After 48 h of treatment, cells were washed twice with PBS, trypsinized and submitted to a hypotonic shock using a solution of RPMI 1640:dH2O:FBS (37.5:12.5:1), and immediately centrifuged. The pellet was spread onto microscope slides that were dried, fixed in absolute methanol (Sigma-Aldrich), stained with 4% Giemsa (Merck, Darmstadt, Germany) and finally air-dried at room temperature. Slides were blind scored under a bright field microscope (Axioskop 2 Plus, Zeiss, Germany) for the presence of micronuclei (MN), using the criteria described by Fenech (2007). At least 2000 binucleated cells (BNC) from two independent cultures were scored per treatment condition and the frequency of micronucleated binucleated cells per 1000 cells (MNBNC/1000 BNC) was determined. The proportion of mono- (MC), bi- (BNC) or multinucleated-cells (MTC) was calculated by scoring 1000 cells per treatment and the cytokinesis blocked proliferation index (CBPI) was calculated as follows [50]:$$\mathrm{CBPI}=\frac{\mathrm{MC}+2\mathrm{BNC}+3\mathrm{MTC}}{\mathrm{Total}\;\mathrm{cells}}$$

### 2.4. Comet Assay

Following A549/THP-1 co-culture exposure for 3 h and 24 h to each CMNM sample, the cells were harvested, and the comet assay was performed as already described [21]. Ethyl methanesulfonate (EMS, 5 mM, Sigma-Aldrich) with an exposure time of 1 h was used as a positive control. Briefly, the cell suspensions were centrifuged, and the pellets resuspended and embedded in 0.8% low melting point agarose, then spread onto 1% agarose-precoated microscope slides. Slides were subjected to lysis under alkaline conditions for a minimum of 1 h, washed twice with buffer (40 mM HEPES, 100 mM KCl, 0.5 mM EDTA, 0.2 mg/mL BSA, pH 8) and treated either with buffer or with 50 μL of formamidopyrimidine DNA glycosylase (FpG, New England Biolabs, Ipswich, MA, USA), for 30 min, at 37 °C. The slides were then placed into cold electrophoresis buffer for 30 min to allow DNA unwinding under alkaline conditions followed by a 25 min electrophoresis at 0.8 V/cm. Finally, slides were neutralized with PBS, rinsed with distilled water, dried overnight, and stained with ethidium bromide (0.125 μg/μL). In each experiment, a total of 100 randomly selected nucleoids (50 nucleoids per gel) were analysed in FpG-treated and untreated gels from each culture, under a fluorescence microscope (Leica Dm500, Leica Camera, Wetzlar, Germany) using the Comet Assay IV image analysis system (Perceptive Instruments, Cambridge, UK). The percentage of DNA in the tail was chosen as a measure of DNA damage. The results represent the Mean ± Standard Deviation (M ± SD) of the median of at least three independent experiments, each with two replicates per treatment condition.

### 2.5. Statistical Analysis

Normality of data was confirmed with Q–Q percentile plots and Kolmogorov–Smirnov tests. Equality of variances was evaluated using Levene’s test. Statistical comparisons of comet assays data between treated and control cells were performed through one-way analysis of variance (ANOVA) followed by Tukey’s multiple comparison test, after testing for the data normality, or the Kruskal–Wallis test when normality was not observed. The two-tailed Student’s t-test was used to compare the results obtained with and without FpG treatment, and to compare the CBPI results between the treated and control cells. The 2-tailed Fisher’s exact test was applied to analyze the results of the frequency of micronucleated cells between exposed and non-exposed cells. Statistical significance was set at *p* < 0.05. All analyses were performed with the SPSS statistical package (version 22, SPSS Inc. Chicago, IL, USA).

## 3. Results

The potential genotoxicity of the three CMNM samples was evaluated by the micronucleus and comet assays in the aforementioned co-culture of A549 and THP-1 cells. The exposure of the co-culture to each CMNM did not produce significant alterations in the frequency of MNBNC, as compared to the controls (Figure 2). Moreover, the cytokinesis-block proliferation index (CBPI) of A549 cells was not affected by exposure to the CMNMs (Figure 2). The CBPI indicates the average number of cell cycles per cell during the period of exposure, thus, these results indicate no effect of CMNMs in cell cycle progression.

Regarding the comet assay results, only the lowest concentrations (1.5 and 3 µg/cm^2^) of the CMF-ENZ caused a significant increase in the % of DNA damage (assessed by the tail intensity) over the control after 24 h exposure, with the use of FPG. This finding suggests that the observed damage is related with oxidant damage to DNA. Although the % of DNA damage with FPG is no longer significant (*p* > 0.05) for all other concentrations tested as compared to the non-exposed control, the differences between samples treated with and without FPG are still statistically significant in the two following concentrations (6 and 12.5 µg/cm^2^; *p* = 0.000 and *p* < 0.05, respectively). This effect was not observed after 3 h exposure. All other CMNMs did not cause DNA damage, either at 3 h or 24 h exposure (Figure 3).

## 4. Discussion

CMNMs have attracted the interest of industry and biomedicine due to their unique physicochemical properties and multiplicity of applications, increasing the likelihood of human exposure in environmental and occupational settings, through diet or consumer products. This has raised concerns about the potential adverse effects of CMNMs on human health, because if cellulose is a natural and biodegradable material in nature, the evidence of its possible biopersistency in the lungs [19], the nanoscale dimension and the intrinsic properties of NMs may give rise to biological effects that might be close to those reported for other nanomaterials such as carbon nanotubes or graphite. Therefore, it is of utmost importance to evaluate CMNMs toxicity prior to their widespread use, including their genotoxicity. In vitro genotoxicity testing of nanomaterials must detect relevant events that may lead to malignancy, i.e., DNA damage, clastogenicity and aneugenicity, and these are events detected by the combined use of the in vitro cytokinesis-blocked micronucleus (CBMN) assay and the comet assay. Thus, in this study, we have applied these two well-known toxicological assays to analyze three different CMNMs, two micro/nanofibrillated celluloses and a nanocrystalline cellulose. We used a co-culture of epithelial alveolar A549 cells and THP-1 macrophages, since inhalation is one possible main route of human exposure, particularly in occupational settings, and the immune system is a key player in the response to non-soluble nanomaterials. In fact, several in vitro [28,32,51,52,53,54] and in vivo studies [26] have suggested that CMNMs, particularly CNC, can elicit an immunotoxic response, although at a much lower level than other nanomaterials such as carbon nanotubes. Macrophage activation can result in genotoxicity, particularly in situations where chronic inflammation persists generating reactive oxygen species that cause pre-mutagenic DNA lesions [55].

The results from the micronucleus assay showed no effect on CBPI and no significant increase in the frequency of micronucleated cells after 48 h exposure to each CMNM, suggesting no cytotoxic or genotoxic effect, respectively. It is generally assumed that CMNMs are not cytotoxic (reviewed in [24]), although there are studies indicating some cytotoxicity with longer exposure times [29]. In a previous study using the same concentration range of these three CMNMs, none induced a significant cytotoxic effect in A549 cells as compared to the controls, after a 24 h exposure period [47]. Regarding genotoxicity, by contrast, in another study using this same co-culture system and a different CNF-TEMPO sample with a lower yield (82.4%) and carboxyl group content (1177 µeq g/g), and higher degree of polymerization (309) and fiber diameter (18.5 nm), the two lowest CNF-TEMPO concentrations tested were able to increase significantly the frequency of micronuclei, while the highest ones had no effect [21]. In addition, other authors reported no genotoxicity assessed by the cytokinesis-block micronucleus assay and the alkaline comet assay in BEAS-2B cells exposed to 1.6–131.6 µg/cm^2^ of nonfunctionalized, carboxymethylated, phosphorylated, sulfoethylated and hydroxypropyltrimethylammonium-substituted CNF [56]. Likewise, neither cotton-derived CNC (average length 135 ± 5 nm; width 7.3 ± 0.2 nm) nor microcrystalline cellulose, in concentrations ranging from 0.5 to 20 µg/cm^2^ increased the level of DNA damage in BEAS-2B cells exposed for 48 h [57]. In our previous work testing the same CMNM samples in A549 cells, but without the presence of THP-1 cells in culture, CNF-TEMPO and CNC also showed no genotoxicity [47]. Nevertheless, a significant genotoxic effect, as assessed by the micronucleus assay, was observed at the lowest and highest CMF-ENZ concentrations tested (1.5 and 50 µg/cm^2^), which was not reproduced in this co-culture model. This finding suggests that THP-1 cells may be modifying the cellular response of A549 cells to CMF-ENZ treatment. Since both cell types are cultured separately in chambers divided by a filter, this influence of THP-1 cells can only be attained through the diffusion of mediators in the culture medium. In fact, THP1 cells are much more susceptible to cytotoxicity, cellular damage and inflammatory responses after CMNM exposure than A549 cells [53]. Thus, their presence could exacerbate the adverse effects of CMNMs on A549 cells, for instance, through the release of cytokines and other mediators. Moreover, the presence of THP-1 cells in a similar co-culture system was shown to change the genotoxic effects of another nanomaterial, a MWCNT, in A549 cells, an effect associated with the epithelial-mesenchymal transformation (EMT) of A549 cells [39]. EMT is a process that leads epithelial cells to assume a mesenchymal phenotype, which confers apoptosis resistance, enhanced migration and invasiveness, and increased production of extracellular matrix components during tumorigenesis [58]. It was also at the lowest CMF-ENZ concentration tested (1.5 µg/cm^2^) and at the immediately following one (3 µg/cm^2^) that a significant DNA damage level was observed (comet assay). Nevertheless, this happened only after 24 h exposure and upon the action of FPG, a glycosylase that recognizes and cuts oxidized bases, thus originating DNA breaks. These DNA lesions are likely to be mutagenic when the cellular mechanisms of DNA repair fail to correct them, and mutagenesis is implicated in carcinogenesis [59]. In a previous study, we showed that none of these three CMNMs was able to induce reactive oxygen species formation after A549 cells exposure [47], but the presence of macrophage-like cells might have promoted their formation. Other studies were unable to detect oxidative damage with CMNM exposure, although some have detected a small effect by using several oxidative stress markers after exposure to CNF and CNC [24,47]. The other concentrations of CMF-ENZ, as well as the other CMNM samples, did not significantly increase the levels of DNA damage, either at 3 h or at 24 h exposure, even with the addition of FPG. This indicates no genotoxicity for CNF-TEMPO and CNC, as reported previously in bronchoalveolar lavage (BAL) cells from mice exposed to CNF-TEMPO by a single pharyngeal aspiration [26]. However, it also suggests that low concentrations of CMF-ENZ may be critical in terms of genotoxicity. In fact, in previous studies, only the two lowest concentrations tested of the same CNF-TEMPO and CMF-ENZ (1.5 and 3 µg/cm^2^) caused a significant increase in micronuclei frequencies in osteoblastic-like human cells (MG-63 cells). In addition, chinese hamster lung fibroblasts (V79 cells) displayed an increased frequency of micronuclei after treatment with 3 and 12.5 µg/cm^2^ of CNF-TEMPO, being much more resistant to the genotoxic effects of CNMNs than MG-63 cells [29]. The general absence of genotoxicity of CMNMs, particularly at higher concentrations, may be related to their entanglement and aggregation, as already demonstrated [47]. Furthermore, in the protein-rich culture medium, it is likely that a protein corona is formed around the CMNMs, further increasing their size, which would reduce their bioavailability at the nanometric scale. This effect is compatible with the higher diameter of the CMF-ENZ and CNC samples dispersed in complete culture medium (containing all supplements and FBS) as compared to those dispersed in PBS (Table 1). On the other hand, CMF-ENZ is slightly less negatively charged than the other CMNMs under study (Table 1), a fact that might facilitate its interaction with the cell membrane (also negatively charged), as compared with the other CMNMs. Indeed, the CMF-ENZ used in this study was the CMNM that demonstrated higher internalization by A549 cells, being visible within endosomes and in the cell cytoplasm [47]. On the contrary, minimal internalization of CNF-TEMPO and no evidence of CNC internalization was observed [47]. In another study, it was shown that a nanocrystalline cellulose was internalized by BEAS-2B, but only after a long exposure period of 4 weeks [57]. Moreover, negatively charged CNC-FITC was negligibly internalized by eleven different cell lines [60,61,62].

Besides the lack of a dose–effect relationship on the genotoxicity of CMNMs, the studies previously mentioned also highlight cell type-specific responses to the same CMNM, which may be related to differences in cellular capability to uptake CMNMs or in the DNA damage response. In that respect, the use of more advanced in vitro models, such as 3D models, would be also valuable for the genotoxicity assessment and expected to provide further insights on the possible genotoxic effects of CMNMs, contributing to the replacement and reduction of animal testing. Also, different sources of raw materials, isolation and processing/manufacturing procedures, drying methods and surface functionalization may have an impact on the type, size and structure of CMNMs and, consequently, on its genotoxicity [47]. All these variables can result in apparently contradictory findings and make it difficult to compare different studies, leaving some uncertainty about the genotoxicity of CMNMs, similar to what happens with other types of nanomaterials.

## 5. Conclusions

This study indicates that the samples of CNF-TEMPO and CNC under study were not genotoxic under the conditions tested, while low concentrations of the CMF-ENZ sample was able to produce a low level of oxidative DNA damage in lung-derived cells. That damage did not appear to result in more permanent and severe chromosomal alteration that would have been detected by the micronucleus assays. These findings are in line with other toxicological studies and contribute to the weight of evidence on the non-genotoxicity of fibrillated and crystalline micro/nanocelluloses. However, they have still to be interpreted with caution, avoiding generalization, because CMNMs with different physicochemical properties may produce different (or adverse) effects. Thus, more research needs to be done, preferably with low doses and longer or repeated exposure conditions, which better reflect real-life human exposure. The use of co-cultures, as the A549-THP1 co-culture here applied, also better mimics the in vivo complexity of the lung and its use should be promoted as a more advanced cell system for nanotoxicological studies.

## Figures and Tables

**Figure 1 bioengineering-10-00986-f001:**
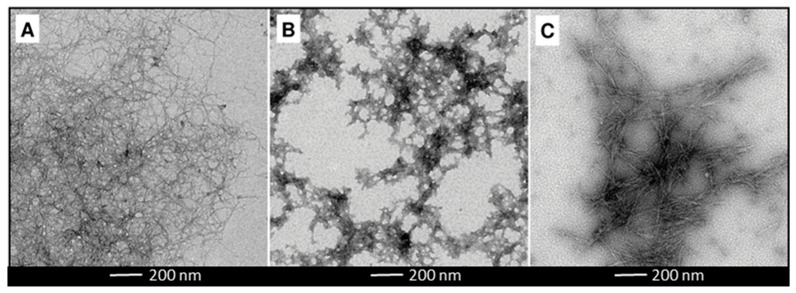
TEM images of the cellulose micro/nanofibrils dispersed in PBS. (**A**) TEMPO-mediated oxidation, (**B**) enzymatic pre-treatment and (**C**) cellulose nanocrystals (calibration bar = 200 nm).

**Figure 2 bioengineering-10-00986-f002:**
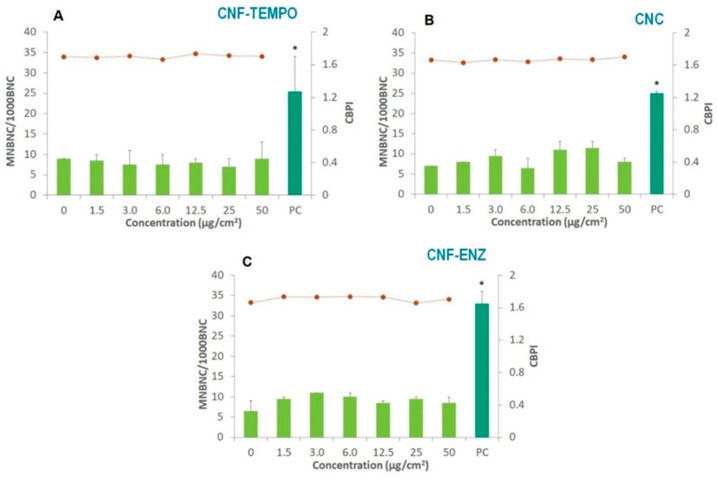
Results of the micronucleus assay in A549 cells co-cultured with THP-1 cells after exposure to (**A**) CNF-TEMPO; (**B**) CNC and (**C**) CNF-ENZ. Light green bars represent the frequency of micronucleated binucleated cells per 1000 binucleated cells (MNBNC/1000 BNC); the dotted line represents the cytokinesis-blocked proliferation index (CBPI). The dark green bar represents the frequency of MNBNC/1000 BNC for mitomycin C, used as the positive control (PC). Results are expressed as M ± SD. * *p* < 0.0001.

**Figure 3 bioengineering-10-00986-f003:**
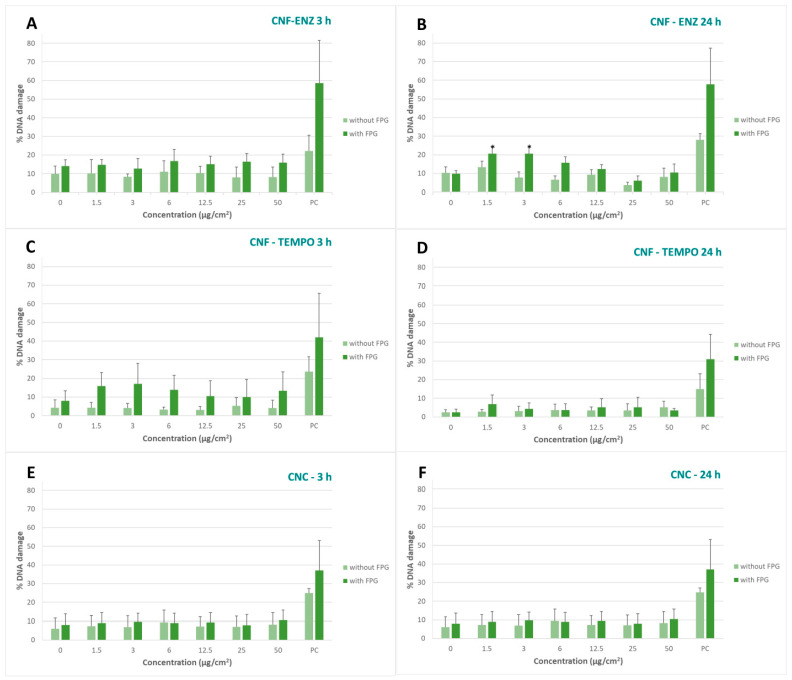
Results of the comet assay after 3 h and 24 h of exposure to (**A**,**B**) CNF-ENZ, (**C**,**D**) CNF-TEMPO and (**E**,**F**) CNC, respectively, in A549 cells co-cultured with THP-1 cells. Columns represent the percentage of DNA damage, lighter green without FPG and darker green with FPG. EMS was used as the positive control (PC). Results are expressed as M ± SD. * *p* < 0.05.

**Table 1 bioengineering-10-00986-t001:** Properties of the cellulose nanomaterials under study, previously reported by our group [47].

Nanocellulose Sample	Yield (%)	C_COOH_ (μmoL/g)	DP	[η] (mL/g)	Fibril Diameter ^1^ (nm)		z-Potential (mV)	
					PBS	CM	PBS	CM
CNF-TEMPO	100	1332	309	130	10.7 ± 1.9	-	−24.6 ± 1.0	−19.7 ± 1.5
CMF-ENZ	4.9	143	1591	618	29.7 ± 7.3	85.2 ± 41.2	−11.6 ± 1.0	−9.4 ± 0.6
CNC	-	-	-	-	19.7 ± 6.1	36.0 ± 9.0	−17.3 ± 0.8	−13.9 ± 0.3

C_COOH_: Carboxyl group content; DP: Degree of polymerization; [η]: Intrinsic viscosity; CM: RPMI cell culture medium; ^1^: Estimated by TEM imaging.

## Data Availability

The data supporting the reported results are available upon request from the corresponding author.

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
