# Peer review of "Assessing the Genotoxicity of Cellulose Nanomaterials in a Co-Culture of Human Lung Epithelial Cells and Monocyte-Derived Macrophages"

_bioengineering, 2023, doi:10.3390/bioengineering10080986_

Round 1

Reviewer 1 Report

Nanocelluloses have a wide number of applications. However, in recent years there has been a wide variety of micro/nano plastic pollution associated with cellulose-based materials. The current paper attempted to study the effect of such on the cells. The studies described by the authors are highly organized and well-written. 

There are some comments which are minor and need to be added in the revision

1. How author choose to control for performing such an experiment and what was the rationality (if considered), please clarify as in the current form it is a bit confusing

2. Was there any studies performed by the authors in relation to the size/form of these CMNM with respective to their cytotoxicity, OR "studies considering the surface properties of CMNM with respect to their toxicity profile? "

3. Rationality behind choosing Carboxyl group content (CCOOH), was there any obvious reason?

Author Response

We thank the reviewer for the willingness to review our work and for his/her pertinent questions, which we hope to sucessfully clarify:

  1. How author choose to control for performing such an experiment and what was the rationality (if considered), please clarify as in the current form it is a bit confusing

R: The authors are not quite sure to have understood the question. Regarding the rationale for performing the studies presented, as human exposure to CMNM is growing due to their application in multiple industrial and biomedical fields, our goal was to contribute to ensure their safety by assessing the genotoxicity of CMNM with different physicochemical properties. We used a co-culture of epithelial alveolar A549 cells and THP-1 macrophages, since inhalation is a predicted main route of human exposure in occupational settings, and the immune system is a key player in the response to non-soluble nanomaterials (line 275-279). We had successfully used this co-culture system to study the genotoxicity of multi-walled carbon nanotubes (https://doi.org/10.1080/17435390.2019.1695975), and now we are applying it to the genotoxicity assessment of CMNM. For that, we used two well-known toxicological assays “often used in combination because the comet assay detects the DNA damage/stand breaks and oxidant damage to DNA (FPG-comet assay) with high sensitivity, whereas the MN assay identifies chromosomal breaks or loss, thus characterizing biological events that are linked to cell transformation and cancer [44-46].” (line 123-125). The comet assay was performed three times, each using two replicates; the micronucleus assay was performed in duplicate, each with two replicates, as recommended in the OECD 487 guideline. All assays included a “solvent control”:  PBS and a positive control: comet assay – ethyl methanesulfonate, EMS (5 mM for 1h); micronucleus assay - mitomycin C (50 μg/mL for 6h). This information is included in the materials and methods section and the results are presented in Figures 1 and 2. These controls were selected based on their mechanism of action, literature data, and the existence of historical data obtained in our lab for these compounds.

  1. Was there any studies performed by the authors in relation to the size/form of these CMNM with respective to their cytotoxicity, OR "studies considering the surface properties of CMNM with respect to their toxicity profile? "

R: Thank you for the question. As mentioned in page 8 of the manuscript, “It is generally assumed that CMNM are not cytotoxic (reviewed in [24]), although there are studies indicating some cytotoxicity with longer exposure times [29]. In a previous study using the same concentration range of these three CNMN, none induced a significant cytotoxic effect in A549 cells as compared to the controls, after a 24 h exposure period [47].”

Most articles studying different types of CMNM, including CMNM with different surface funtionalizations, reported that they are not cytotoxic. In our experience with these three different types of CMNM, two fibrillar (one nanometric, and the other mostly micrometric) and one crystalline, only the crystalline CMNM seemed to be cytotoxic after 17 hours of exposure and 4 days resting, as mentioned in the cited manuscript (https://doi.org/10.3390/jox12020009).

  1. Rationality behind choosing Carboxyl group content (CCOOH), was there any obvious reason?

R: Carboxyl groups are important components of bleached cellulosic pulp, but they are significantly increased in catalytic oxidation with 2,2,6,6-tetramethylpiperidine-1-oxyl radical (TEMPO). TEMPO- mediated oxidation  under aqueous conditions has been developed as a pre-treatment of plant cellulose fibres to efficiently prepare nanocellulose (Isogai et al. 2011; Saito et al. 2006; Zhou et al. 2018) while reducing the grinding cycles prior to the vigorous mechanical fibrillation by homogenization or microfuidization. TEMPO catalyzes the oxidation of primary alcohol groups in aqueous media, and regenerated cellulose is converted into water-soluble polyglucuronic acid. Significant amounts of C6 carboxylate groups are selectively formed on each cellulose microfibril surface by TEMPO-mediated oxidation without any changes to the original crystallinity or crystal width of wood celluloses. Electrostatic repulsion and/or osmotic effects working between anionically-charged cellulose microfibrils cause the formation of completely individualized nanofibers dispersed in water by gentle mechanical disintegration treatment of TEMPO-oxidized wood cellulose fibers. This TEMPO-mediated oxidation demonstrated to be more advantageous than conventional chemical reactions used for introducing carboxyl groups onto cellulose. Therefore, the carboxyl group content of cellulose is related to its fibrillation yield, and to the production of nanofibers instead of microfibers.

Reviewer 2 Report

The aim of this work was to evaluate the genotoxicity of cellulose micro/nanomaterials prepared from industrially bleached Eucalyptus globulus kraft pulp. The micronucleus assay was used to detect chromosome breaks or losses, and the comet assay to detect DNA damage/stand breaks and oxidative DNA damage. The assays, which were performed in a co-culture system using human lung epithelial cells and monocyte-derived macrophages, show that the CNF and CNC tested are not genotoxic, while the CMF has a low genotoxic property.

The research results are relevant to the safe use of cellulose nanomaterials in numerous applications. The experiments are well performed and clearly presented, and the conclusions are supported by the results.
I recommend the manuscript for publication in Bioengineering.

Author Response

The authors are grateful for your willingness to review our article and your kind words of acknowledging our work.

Reviewer 3 Report

In the present study, “Assessing the Genotoxicity of Cellulose Nanomaterials in a Co-Culture of Human Lung Epithelial Cells and Monocyte-Derived Macrophages, the authors aimed to assess the genotoxic effects of CMNM produced from industrial bleached Eucalyptus globulus kraft pulp, using the micronucleus assay and the comet assay.

It is known that the primary characterization of nanoparticles is fundamental to associate the results of their potential genotoxic effect with a real DNA damage. For longer exposure times, or for higher concentrations, this damage could be repaired by the mechanisms put in place by the cell itself or an agglomeration of molecules could occur, which in fact has no effect on the DNA. In both cases, therefore, it is important to repeat the characterization for a different experimental model (co-culture of human lung epithelial cells and monocyte-derived macrophages) than the one reported in paper 47 by Pinto et al. Furthermore, it would also be useful to show TEM images of the results obtained about the morphology and estimated diameter.

Given the great interest in the topics, this work could provide important reference information that are of good relevance, but I do not recommend the publication in its present form.

Specific comment

·         The resolution of fig. 2 is very low

Author Response

We thank the reviewer for the wilingness to review our work and for his/her good suggestions to improve it. Our point-by-point response follows below:

In the present study, “Assessing the Genotoxicity of Cellulose Nanomaterials in a Co-Culture of Human Lung Epithelial Cells and Monocyte-Derived Macrophages, the authors aimed to assess the genotoxic effects of CMNM produced from industrial bleached Eucalyptus globulus kraft pulp, using the micronucleus assay and the comet assay.

It is known that the primary characterization of nanoparticles is fundamental to associate the results of their potential genotoxic effect with a real DNA damage. For longer exposure times, or for higher concentrations, this damage could be repaired by the mechanisms put in place by the cell itself or an agglomeration of molecules could occur, which in fact has no effect on the DNA. In both cases, therefore, it is important to repeat the characterization for a different experimental model (co-culture of human lung epithelial cells and monocyte-derived macrophages) than the one reported in paper 47 by Pinto et al. Furthermore, it would also be useful to show TEM images of the results obtained about the morphology and estimated diameter.

Given the great interest in the topics, this work could provide important reference information that are of good relevance, but I do not recommend the publication in its present form.

R: The authors completely agree with the reviewer that the physicochemical characterization of the nanomaterials under study is of the utmost importance to the interpretation of their toxicological effects, and only with a good characterization, a grouping strategy approach to the development of predictive toxicology, as well as a safe-by-design approach can be implemented. Thus, the CMNM used in this study were fully characterized before and in parallel with the toxicological assays, including by TEM, both dispersed in distilled water and in culture medium. The results from this complete characterization were reported by Pinto et al. (2022), since our analysis of the results included in that paper (in the monoculture model) ended first, but the experimental work performed in the present study (co-culture model of A549/THP1) and in Pinto et al. (2022) was performed simultaneously, using the same CMNM. The culture medium used in the monoculture and in the co-culture is the same (RPMI) and the experimental conditions were all maintained the same. In fact, our initial plan was to report the toxicological findings of both models in the same publication in order to better compare them, but data analysis of the results obtained in the co-culture system was delayed due to the end of a post-doctoral research grant, and only now we are reporting that part of the work. Therefore, the CMNM characterization, although published earlier, also applies to the present study. Nevertheless, we completely agree with the reviewer that TEM images of the CMNM would improve our work and, thus, included new images different from the ones that were included in Pinto et al. (2022) in the present manuscript.

Specific comment

  • The resolution of fig. 2 is very low

R: We improved the resolution of that image, thank you.

Round 2

Reviewer 3 Report

The authors revised the manuscript and improved its quality. It can be now considered for publication.

Author Response

Thank you very much for your appreciation.

Yours sincerely,

Maria Silva